# Structure and function of the EA1 surface layer of *Bacillus anthracis*

Adrià Sogues [1,2] ✉, Antonella Fioravanti [1,2], Wim Jonckheere[1,2], Els Pardon [2,3], Jan Steyaert [2,3] & Han Remaut [1,2] ✉

The Gram-positive spore-forming bacterium *Bacillus anthracis* is the causative agent of anthrax, a deadly disease mostly affecting wildlife and livestock, as well as representing a bioterrorism threat. Its cell surface is covered by the mutually exclusive S-layers Sap and EA1, found in early and late growth phases, respectively. Here we report the nanobody-based structural characterization of EA1 and its native lattice contacts. The EA1 assembly domain consists of 6 immunoglobulin-like domains, where three calcium-binding sites structure interdomain contacts that allow monomers to adopt their assembly-competent conformation. Nanobody-induced depolymerization of EA1 S-layers results in surface defects, membrane blebbing and cell lysis under hypotonic conditions, indicating that S-layers provide additional mechanical stability to the cell wall. Taken together, we report a complete model of the EA1 S-layer and present a set of nanobodies that may have therapeutic potential against *Bacillus anthracis*.

Surface layers or S-layers are composed of secreted bacterial (glyco) proteins that readily self-assemble at the cell surface into two-dimensional para-crystalline monolayers that cover the entirety of the producing cells[1]. S-layers have been observed in species of nearly every taxonomical group of bacteria, but despite a shared characteristic to bind the cell surface and form semi-porous lattices, they represent homoplastic structures[1]. The function of S-layers is often poorly understood and seemingly equally diverse, with attributed or proposed roles as a molecular sieve and protective shield, as an adhesin and bacterial virulence factor, or as a cell envelope support structure or "exoskeleton"[2–5].

The Gram-positive, aerobic and spore-forming bacterium *Bacillus anthracis* is the causative agent of the anthrax disease. Despite its low morbidity in humans, anthrax remains a concern for public health due to its high mortality and the lack of effective vaccination and treatment options[6]. Particularly so in light of its potential use as a bioweapon and the resilience of its spores in the environment[7]. *B. anthracis* spores present a complex envelope structure with several protective layers that allow persistence in harsh environments

for long periods of time[8]. Vegetative forms of *B. anthracis* also show a complex cell surface architecture composed of a plasma membrane, an approximately 40 nm-thick peptidoglycan sacculus and an S-layer as proteinaceous monolayer representing the outermost components of the cell envelope[9]. In fully virulent strains, a polyglutamate capsule is present, attached to the PG sacculus and protruding through the semiporous S-layer to cover the cell surface in an anti-phagocytic shield[10–12]. While most organisms only contain a single SLP, the S-layer of *B. anthracis* is dynamic in its composition, presenting two mutually exclusive S-layers made of surface array protein (Sap) or extractable antigen 1 (EA1) appearing at the cell surface during exponential and stationary growth phases, respectively[13]. The functional implications and reasons behind such metabolically costly switch towards an EA1 S-layer during the stationary stage are unknown. Though Sap and EA1 have been implicated in *B. anthracis* virulence[14,15], infection studies on the fate of EA1 or Sap null mutants are lacking. Antibodies against Sap and EA1 were found in convalescent serum samples from individuals who had contracted cutaneous anthrax and developed severe symptoms, indicating that

---

[1]Structural and Molecular Microbiology, VIB-VUB Center for Structural Biology, VIB, Pleinlaan 2, 1050 Brussels, Belgium. [2]Structural Biology Brussels, Vrije Universiteit Brussel, VUB, Pleinlaan 2, 1050 Brussels, Belgium. [3]VIB-VUB Center for Structural Biology, VIB, Brussels, Belgium. ✉e-mail: Adria.Sogues.Castrejon@vub.be; Han.Remaut@vub.be

both S-layer proteins are expressed during infection and accessible to the immune system[16]. Uchida and co-workers found that the EA1 protein remains detectable on the surface of *B. anthracis* endospores and that mice immunized intranasally with recombinant EA1 presented a decreased mortality rate when challenged with anthrax spores[15], an effect not observed in mice immunized with Sap[14].

Both Sap and EA1 proteins contain an N-terminal signal peptide that directs the protein towards the cell surface. The N-terminus of the mature domain starts with a ~180 amino acid region (31–213, Fig. 1a) that comprises three S-layer homology regions (SLH) that anchor the protein to the cell surface through binding of the ketal-pyruvylated *N*-acetylmannosamine unit in the secondary cell wall polysaccharide (SWCP) via non-covalent interactions[17,18]. The remaining part of the protein (214–862, Fig. 1a) forms the assembly domains (AD) or "crystallization domain," responsible and sufficient for self-assembly of the S-layer lattice[14]. Other than the SLH domains, the Sap and EA1 ADs share a mere 22% sequence identity, and assemble into morphologically different S-layer lattices[19,20]. Recently, the structure of the monomeric Sap[AD] was determined using nanobodies as crystallization aids[14]. Sap[AD] comprises a string of six Ig-like domains that fold independently of calcium, a divalent metal that is commonly found to stabilize SLPs and the S-layer lattice[21–24]. Here, we report the X-ray structure of the EA1 assembly domain at 1.8 Å in complex with two EA1-biding nanobodies that inhibit EA1 self-assembly and depolymerize existing EA1 S-layer lattices. The X-ray structure shows the EA1 architecture comprises six Ig-like domains, and reveals 3 calcium-binding sites that structure interdomain contact loops and thereby help EA1[AD] adopt its assembly-competent supertertiary structure. The crystal structure also captures in-plane intermolecular contacts that allow unambiguously docking of the EA1[AD] into a 3D cryo-EM map of reconstituted EA1 S-layers, resulting in a comprehensive model of the EA1 S-layer.

## Results

### Recombinant expression and in vitro reconstitution of the EA1 S-layer

The exponential growth S-layer protein Sap consists of six Ig-like domains connected by short linkers that fold into a "body" formed by domains D3 to D6 and a protruding "arm" composed of D1 and D2[14]. Prediction of the EA1[AD] structure (i.e., corresponding to residues 213 to 862, lacking the SLH domains) using AlphaFold2[25] suggested a condensed structure consisting of six consecutive β-sandwich domains, albeit some with low confidence scores (Supplementary Fig. 1). This predicted EA1 domain topology is reminiscent of that of Sap, composed by an N-terminal signal peptide, followed by three SLH domains and the assembly domain composed of six β-sandwich domains connected by short linkers (Fig. 1a). To provide experimental structural data, we first optimized EA1 purification in the recombinant system *Escherichia coli*. Cytoplasmic expression of EA1[FL] (full-length) or EA1[AD] yielded high levels of protein (~1.5 and 1.81 mg EA1[FL] or EA1[AD]/g cell pellet, respectively). However, recombinant EA1 (both AD and FL) largely phase-separated into a gel-like pellet consisting of a conglomerate of 2D sheets, suggesting the self-assembling nature of EA1 readily led to the in vitro and in cellulo formation of S-layer-like structures (Supplementary Fig. 2a–c). Ni affinity and size exclusion purification of 6xHis-tagged EA1 and EA1[AD] under denaturing conditions (i.e., 8 M urea; see "Methods") allowed us to obtain monodisperse, monomeric samples for the full-length protein and the EA1 assembly domain (Fig. 1b). Upon removal of the chaotropic agent, the purified samples readily adopted a high β-sheet content (Supplementary Fig. 2d) and when concentrated above 2 mg/ml, purified EA1 and EA1[AD] assembled into S-layer-like sheets, suggesting the proteins refolded into their native, assembly-competent conformation (Fig. 1c). Recombinant EA1 and EA1[AD] assembled in two-dimensional lattices with P1 symmetry and unit-cell parameters of $\alpha = 73.8$ Å, $\beta = 87.8$ Å, $\gamma = 107°$ (Supplementary

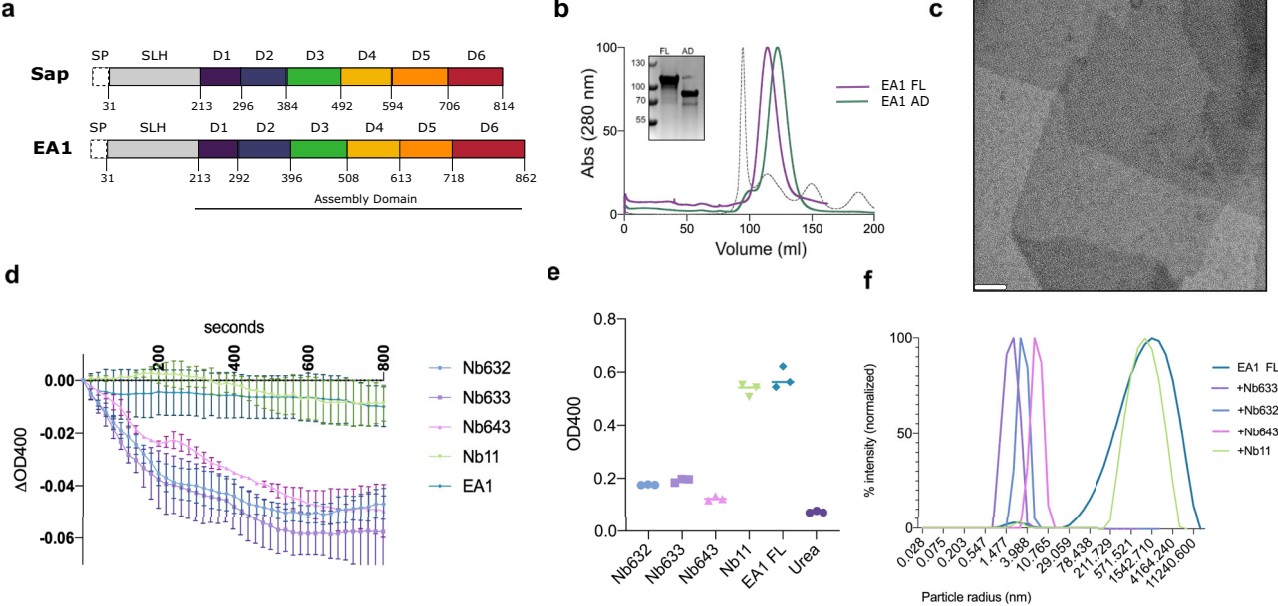

**Fig. 1 | Biochemical characterization of EA1 and depolymerizing nanobodies.**
**a** Schematic representation of Sap and EA1 domains. Both proteins contain a signal peptide (SP) at the N-ter followed by the SLH domain that allows binding to the SCWP decorating the peptidoglycan. The assembly domain is shown in colors and domain boundaries are indicated. **b** Size exclusion chromatography profile of EA1[FL] and EA1[AD] showing a single peak at 280 nm that corresponds to the EA1 monomer. The dotted gray curve represents the ladder with peaks from left to right of 670 kDa, 158 kDa, 44 kDa and 17 kDa. Inset: SDS-PAGE of purified EA1 AD and FL. **c** Negative-staining TEM micrograph of in vitro reconstituted EA1 S-layer after purification and refolding. EA1 forms two-dimensional monolayers that can reach μm in size. The scale bar is 100 nm. **d** Depolymerization assay on EA1 S-layers by nanobodies, decrease in absorbance at 400 nm over time indicates depolymerization. Nb11 is unrelated to EA1. Error bars represent the mean ± SD. **e** Measurement of OD$_{400}$ after 1 h of incubation with nanobodies. **f** EA1[FL] particle size distribution by DLS in the presence of different nanobodies. Measurements were taken after 24 h and indicated a total absence of S-layers. The data shown are representative of experiments made independently in triplicate.

Fig. 3). These values closely resemble the lattice parameters previously reported for EA1 S-layers expelled from *B. anthracis* cells ($\alpha = 69$ Å, $\beta = 83$ Å, $\gamma = 106°$)[26], indicating that in vitro reconstructed S-layer assembles into the same lattice as found on the bacterium's surface.

## EA1 nanobodies depolymerize existing EA1 S-layers in vitro

Nanobodies are single-domain antibodies derived from camelid species that can be used as crystallization chaperones[27]. Previous work showed that nanobodies can prevent, even revert, S-layer assembly and allow 3D crystallization and structure determination of SLPs[21,14]. We therefore raised nanobodies against the assembly domain of EA1 and selected sixteen unique EA1-binding nanobodies belonging to seven different families. We screened for nanobodies with depolymerizing activity by incubating them with pre-assembled EA1 S-layers and following the turbidity at 400 nm over time. Three of the nanobodies (Nb[632], Nb[633] and Nb[643]; Supplementary Fig. 4), but not a control Nb (i.e., Nb11, binding an unrelated target)[14], induced a gradual reduction in optical density over a 60 min period (Fig. 1d), reaching an absorbance similar to that of the sample treated with 8 M urea after 1 h (Fig. 1e). Negative-stain transmission electron microscopy (TEM) and dynamic light scattering (DLS) showed the S-layers depolymerized to monomeric EA1 (Supplementary Fig. 5 and Fig. 1e). Bio-layer interferometry reported that Nb[633], Nb[632] and Nb[643] bind to EA1 with a $K_d$ value of 13, 2.2, and 1.9 nM respectively, values that are in the expected range for a nanobody–protein interaction (Supplementary Fig. 6).

## Atomic details of the EA1 monomer in complex with Nb[632] and Nb[643]

Most SLPs form 2D crystals but fail to crystallize in a third dimension, hampering structure determination by X-ray crystallography. We screened for the 3D crystallization of EA1[AD] monomers in complex with single Nbs or combinations of Nb[632], Nb[633] or Nb[643]. The combination of Nb[643] and Nb[632] gave rise to plate-shaped crystals that diffracted to 1.81 Å resolution (Supplementary table 1). The crystal structure revealed that the EA1[AD] consists of six distinct Ig-like domains (D1–D6; Supplementary Fig. 7a, b) connected by short linkers (Fig. 2a). D1 to D6 are arranged forming an overall R-shaped quaternary structure of 70 Å by 115 Å wide and 27 Å thick (Fig. 2b, c). Domains D3-D6 lie in a single plane and arrange into a flat "body," forming a circular interdomain arrangement with a central cavity of 45 Å diameter (Fig. 2). D1 and D2 arrange in a 54.5° angle and form a two-domain "arm" that is connected via a flexible linker (residues 394–400) to the D3-D6 EA1[AD] body. This arrangement in a D1–D2 arm and a D3–D6 body is equivalent, though not isomorphous to the domain topology seen in Sap (Supplementary Fig. 7c) and SbsB, one of the five S-layer proteins encoded in thermophilic bacterium *Geobacillus stearothermophilus* (Supplementary Fig. 7d). Nb[643] and Nb[632] bind the hinge between D1 and D2, and the exposed face of the D2 β-sheet, respectively (Fig. 2a). Thus, since Nb[632] and Nb[643] both have an S-layer depolymerizing activity, at least the local conformation and/or steric accessibility of D1 and D2 are likely to differ from that in the EA1[AD] conformation found in the S-layer (see below). In SbsB, discrete calcium binding sites in the D3–D6 body control the organization of these domains into their assembly competent conformation (Supplementary Fig. 7d)[21]. Interestingly, the X-ray structure of EA1 presents strong electron density and plausible coordination chemistry for three candidate calcium ions, located in D3, D4 and D6 (Fig. 2d–f and Supplementary Fig. 8a). The X-ray fluorescent spectrum for the EA1 crystals showed K-edge emission peaks for Ca, Fe, Cu, and Zn (Supplementary Fig. 8b). Data collection above the Fe, Cu or Zn K-edge absorption peaks did not result in an anomalous EA1[AD] diffraction signal, identifying these elements as contaminants from the cryoloop pin (Fe, Cu) and/or solution (Zn), and identifying the bound metals as Ca$^{2+}$, in agreement with the coordination geometries of the binding sites. A first calcium ion (Ca-1) is located in D3, in a pentagonal bipyramidal ligation with residues in an extended loop (i.e., D411, D413,

D415, V417, and D428 in interface loop 1 or "IL1," residues 411 to 434) located in the interface with D4 (Fig. 2d). A second calcium ion (Ca-2) is located at the outer edge of D2 in pentagonal bipyramidal ligation with an extended loop (i.e., residues T548, I551, E553 and D567, in "IL2," residues 539 to 551 and in "IL3, residues 555 to 577; Fig. 2e). Finally, a third calcium ion (Ca-3) is in octahedral coordination by a short loop (i.e., through residues D784, N786, D788, V790, N792 and D795 in "IL4," residues 785 to 795) near the N-terminal region of D6, in contact with D3 (Fig. 2f). Strikingly, all three calcium-binding sites encompass loop regions located in an interdomain interface, i.e., IL-1 in D3–D4, IL-2 and IL-3 in D4–D3 and IL-4 in D6–D3 (Fig. 2f). Similar to what is seen in SbsB, calcium binding to the EA1[AD] can be expected to exert a structuring and stabilizing activity on these interface loops, and therefore the adoption of the assembly competent quaternary structure of the D3-D6 body. In support of this, we find EA1 monomers are unable to assemble into in vitro S-layers in the presence of the divalent metal chelator EDTA during refolding (Supplementary Fig. 8c).

## Lattice model of the EA1 S-layer

Since EA1 self-assembles into S-layers, and the D3-D6 body showed a Ca$^{2+}$ induced condensation into a flat circular arrangement, we wondered if the crystal structure captured any of the S-layer lattice contacts. The crystal packing shows the presence of stacked planes of EA1[AD] protomers (Supplementary Fig. 9) related by translational vectors $a = 74.3$ Å, $b = 87.7$ Å and $\gamma = 107.8°$, values identical to those that we measured on single reconstituted S-layers imaged by cryo-electron microscopy (Supplementary Fig. 3). To gain further insights into the arrangement of EA1 within the S-layer, we collected cryo-EM data on in vitro reconstituted S-layers (Fig. 3a). 2D classes featured a cruciform pattern similar to that observed in the two-dimensional lattice of the crystal (Fig. 3b and Supplementary Fig. 10a). Although S-layer sheets showed a strong preferred orientation (Supplementary Fig. 10b), the available random tilt in the particles allowed single-particle cryo-EM processing and 3D reconstruction resulting in a map with global resolution estimate of 6.61 Å (Supplementary Fig. 10c). Despite residual missing wedge artefacts along the Z-axis this map allowed unambiguous docking of the EA1[AD] D3-D6 body as observed in the X-ray structure (Fig. 3c). The X-ray structure and 3D cryoEM map show a close match for the D3–D6 body (correlation coefficient of 0.71 as measured with the function fit-in-map in ChimeraX suite). No similar match was found for the D1–D2 arm likely as a result of their interaction with the S-layer disrupting nanobodies Nb[632] and Nb[643]. However, the cryo-EM map showed an unfilled density in plane with the D3–D6 body, seemingly continuous to the N-terminus of domain 3 (Fig. 3c), and wedged between domain 5 and domain 4 of two adjacent EA1 protomers. A rigid body docking of D2 by reorientation around the D2–D3 hinge region as found in the EA1[AD] X-ray structure provided a good match of D2 and the vacant density in the map (i.e., CC of 0.66; Fig. 3d).

Thus, a model emerges where the EA1 S-layer encompasses four intermolecular contact zones (Fig. 3e): (i) D2 and D4$_{BR(bottom\ right)}$ (surface area of 285.4 Å), (ii) D2 and D5$_{BL(bottom\ left)}$ (surface area of 334.6 Å), (iii) D3 and D6$_{R(right)}$ (surface area of 558.7 Å), and (iv) the linker contacting D4 and D5 with D6$_{TR(top\ right)}$ (surface area of 590.1 Å). The EA1[AD] X-ray lattice captures two out of four contact zones (i.e., (iii) and (iv)), providing atomic detail of the binding surfaces (Fig. 4a, b). Of these, the D4-5 and D6$_{TR}$ interface (i.e., interface iv) forms the most extensive interaction, encompassing 5 main chains ($^{MC}$) and side chain ($^{SC}$) hydrogen bonds (G613$^{MC}$–F734$^{MC}$; N614$^{SC}$–N733$^{MC}$; V615$^{MC}$–N858$^{MC}$ and L618$^{SC}$–P869$^{MC}$) and two salt bridges: E612 (D4) and K737 (D6$_{TR}$) and K702 (D5) with the D6$_{TR}$ C-terminal carboxylate in L862 (Fig. 4a). A second lattice contact (i.e., iii) provides a tight packing between the outer β sheet of the D3 and D6, involving five H-bonds (A485$^{MC}$–E846$^{SC}$; K835$^{SC}$–E483$^{MC}$, E483$^{SC}$–S807$^{MC}$, S807$^{SC}$–S474$^{SC}$, E483$^{SC}$–S807$^{SC}$), a salt bridge: H470 (D3) and D808 (D6), and hydrophobic contacts (Fig. 4b).

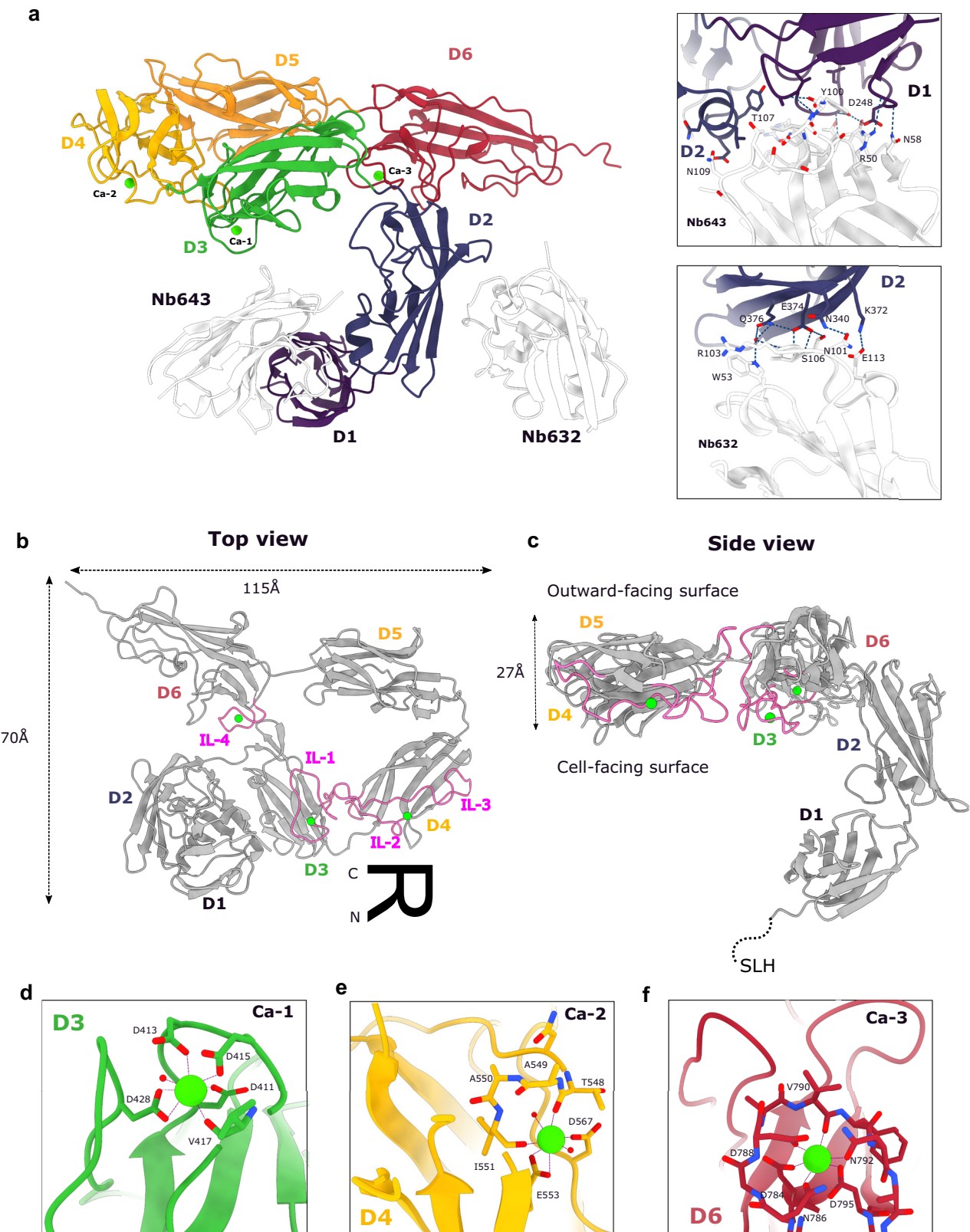

It is worth noting that the calcium-interacting loops in EA1 are not directly involved in intermolecular contacts. Rather, they direct the relative domain orientation within the D3-D6 body, likely facilitating its condensation into the assembly-competent supertertiary structure of EA1. To evaluate if the lattice present in the X-ray structure captures physiologically representative S-layer contacts, we introduced two sterically bulky point mutations (EA1$_{N614W-A484Q}$) in N614 and A184,

which are found in the lattice interface of domains 5 and 3, respectively. In a second mutant, we deleted domain 6 (EA1$_{\Delta D6}$), a domain found at the intersection of several EA1 lattice contacts (Supplementary Fig. 9). We used DLS to assess the ability of these mutants to form S-layers. While EA1 WT showed a wide distribution of high particle radii, characteristic of protein polymers, both EA1$_{N614W-A484Q}$ and EA1$_{\Delta D6}$ present a single peak at a radius expected for EA1 monomer

**Fig. 2 | X-ray structure of EA1$_{AD}$ in complex with nanobodies. a** Overall crystal structure of EA1$_{AD}$ in complex with Nb$^{632}$ and Nb$^{643}$ shown in ribbon representation. Domains are colored differently and domains number are indicated next to it. Nanobodies are shown in white and calcium ions are shown as green spheres. (right top box) Close-up view of the Nb643 binding epitope between D1 and D2. Side chains of contacting residues are shown in stick representation. (right bottom box) Close-up view of the Nb632 binding the exposed β region of D2. Hydrogen bonds and salt bridges are shown as blue dotted lines. **b**, **c** Top and side view of EA1$_{AD}$ shown in ribbon representation. For clarity, nanobodies are not shown. EA1 forms an overall R-shape quaternary structure 70 Å by 115 Å wide and 27 Å thick. Domains 3 to 6 lie in the sample plane while D1 and D2 project out of the sample plane towards the cell wall. Interface loops (IL) are represented in magenta and calcium atoms in green. **d**–**f** Close-up views of the three calcium coordination sites in the EA1 monomer. Calcium ions are found structuring interface-involved loops in D3, D4 and D6 binding Ca-1, Ca-2 and Ca-3 respectively. Metal coordination is represented with purple dotted lines and interacting residues are shown as sticks. Ca-1 and Ca-2 present a pentagonal bipyramid coordination whereas Ca-3 shows a octahedron coordination.

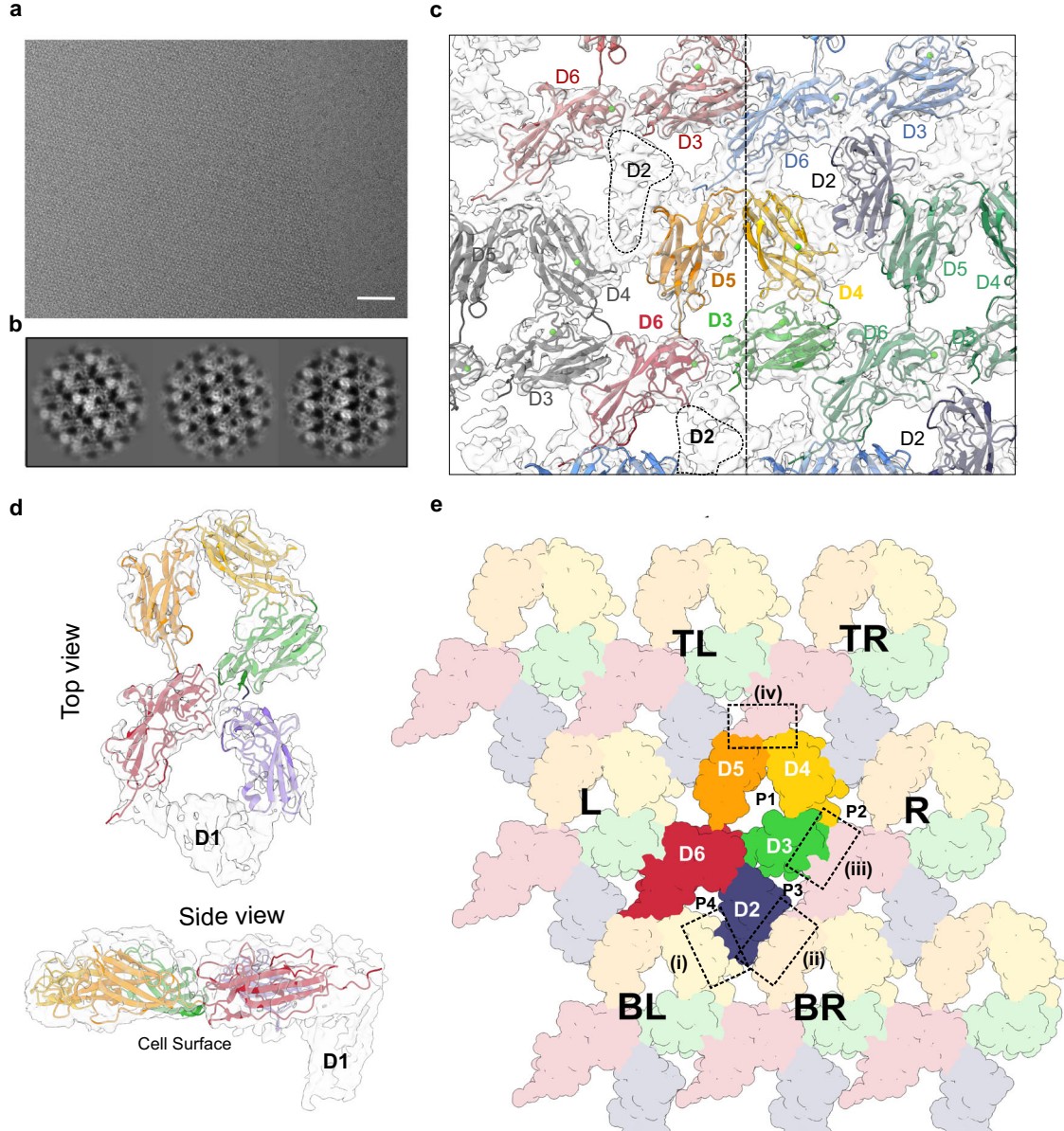

**Fig. 3 | Cryo-EM reconstruction of in vitro reconstituted EA1 S-layers. a** Raw representative cryo-EM image of an EA1 S-layer monolayer. Scale bar is 10 nm. The data shown are representative of experiments made independently in triplicate. **b** Example of two-dimensional class averages of EA1 lattice used for cryo-EM reconstruction. 2D classes featured a cruciform pattern. **c** Top view of the EA1 S-layer with the X-ray lattice (domains from 3 to 6) docked as a rigid body into the cryo-EM density map, unfilled density corresponds to domain 2. Docking of the domain 2 as a rigid body is shown on the left of the dotted line. Domain numbers are indicated in (**d**). Fitting of the EA1 in-plane domains (D2–D6) into the cryo-EM map. Top and side views are shown. Unfilled density adjacent to D2 and protruding downwards belong to D1. **e** Complete EA1 S-layer structure involving D2–D6 represented with all atoms as spheres. Domains are colored differently and domain number is indicated. Central EA1 monomer is colored with vivid colors and neighbor monomers are pale and indicated with letters L left, R right, TR top right, TL top left, BR bottom right, BL bottom left. The EA1 S-layer is stabilized by four intermolecular interaction zones (namely, i, ii, iii, and iv) indicated as dotted squared boxes and forms four pores of different sizes (p1, p2, p3, and p4).

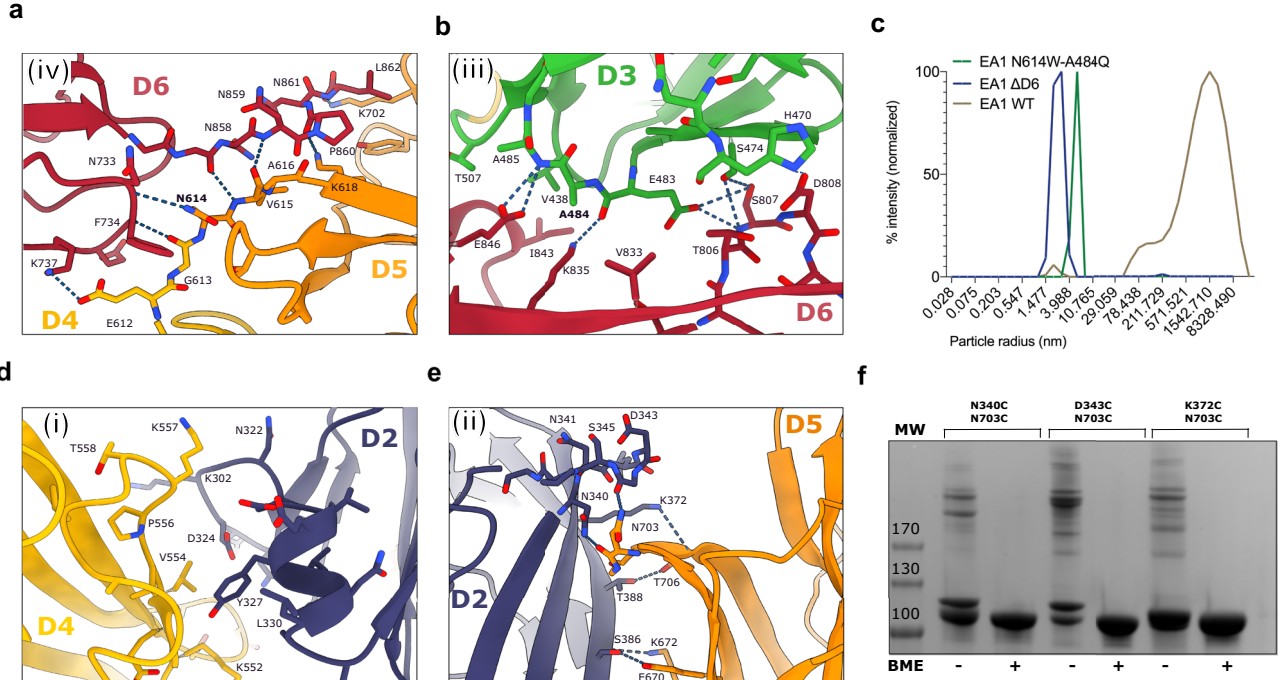

**Fig. 4 | Interface contacts and validation of the EA1 S-layer model. a** Close-up view of the interface (iii) where D6 interacts with D3. Many hydrophobic residues participate in this interface that involves the outer region of the β sheets of D6 with the N-terminal region of the D3. **b** Close-up view of the interface (iv) where the very C-terminal linker of D6 interacts with the linker that connects D5 and D4 of a neighbor EA1 molecule. **c** Particle size distribution was measured by DLS of $EA1_{FL}$, and mutants ($EA1_{N614W-A484Q}$ and $EA1_{\Delta D6}$). Both mutants show a monodisperse single peak at the expected monomer radius size and therefore indicate the absence of an S-layer. **d** Detailed view of the D4 and D5 interface (i) after rigid body docking into the cryo-EM density with RosettaDock-4.0. **e** Close-up view of the interface (ii) where D2 interacts with D5 after rigid body docking into the cryo-EM density with RosettaDock-4.0. Hydrogens bonds are shown as dotted lines. **f** Coomassie-stained SDS-PAGE of reduced (+) and non-reduced (−) EA1 Cys pairs mutants in D2 and D5. Crosslinked EA1 forms covalent bonds in non-reducing conditions between different monomers that lead to very high molecular species bands. In reducing conditions EA1 forms a single band that belongs to the monomer. BME 2-mercaptoetanol, MW molecular weight marker. The data shown are representative of experiments made independently in triplicate.

(Fig. 4c). Thus, surface mutants in the D3-D5 interface are sufficient to thwart S-layer formation, suggesting the interface captured in the X-ray structure reflects native lattice contacts in the EA1 S-layer.

In the S-layer model, the docked D2 domain goes into contact with $D4_{BL}$ (i) and $D5_{BR}$ (ii) (Fig. 3e). RosettaDock-4.0 was run to obtain energy-minimized interfaces of the manually docked models[28]. The D2 and $D4_{BL}$ interface (i.e., interface i) presents the least extensive interaction surface with no apparent electrostatic interactions and few hydrophobic contacts between D2 and $D4_{BL}$ (Fig. 4d). On the other hand, the D2 and $D5_{BR}$ interface (i.e., interface ii) seemingly presents 6 hydrogens bonds between the outer β sheet of the D2 with the outer β sheet of the $D5_{BR}$, involving the residues pairs: $G344^{MC}$-$N703^{SC}$; $K372^{SC}$-$T706^{MC}$; $N340^{SC}$ -$N704^{MC}$; $T388^{SC}$ -$T706^{SC}$; $S386^{SC}$ -$E670^{SC}$; $S386^{MC}$-$K672^{SC}$ (Fig. 4e). To experimentally validate this model we selectively introduced Cys residues in spatially close regions of D2 and D5: N340C, D343C or K372C in D2, each in combination with N703C in D5 (Fig. 4e). These three double Cys mutants were purified, allowed to form S-layers, and analyzed by SDS-PAGE in presence or absence of the reducing agent 2-mercaptoetanol (BME). The three mutants were able to form BME-sensitive high molecular mass species indicating the presence of crosslinked S-layers (Fig. 4f). We used western blot to confirm that the high molecular bands correspond to EA1 (Supplementary Fig. 11a). The fact that the different pairs of Cys mutants are able to form disulfide bridges points to the flexibility of these loops within the S-layer. TEM inspection of the three cysteine mutants confirmed that the high molecular species do not form amorphous aggregates, but rather S-layers with lattice parameters $\alpha = 71$ Å, $\beta = 84$ Å, $\gamma = 107°$ similar to the WT values (Supplementary Fig. 11b, c). Thus, the EA1 S-layer encompasses an in-plane packing of D2-D6 "tiles." In the X-ray structure of $EA1^{AD}$, binding of $Nb^{632}$ positions D2 out of

plane of the D3-D6 body, and overlaps with its binding interface with $D5_{BR}$ in the S-layer, providing a structure-based explanation on the S-layer depolymerizing activity of $Nb^{632}$, and highlighting the importance of the D2 contact for EA1 lattice maintenance. Finally, the cryoEM maps reveal an extra density corresponding to domain 1 (D1) based a good model to map correlation coefficient (CC: 0.72) and its relative positioning continuous to D2 (Supplementary Fig. 12). D1 does not form part of the lattice contacts, but instead protrudes downwards towards the cell wall, projecting the SLH domains in position for SCWP binding. In agreement with this, an EA1 mutant lacking the domain 1 ($EA1_{\Delta D1}$) assembled into large MW particles as judged by DLS, and showed the formation of single-layered 2D crystals in TEM (Supplementary Fig. 13). Taken together, we present a complete model of the EA1 surface layer assembly domain, and its packing in the EA1 S-layer. The EA1 S-layer represents a tightly packed 2D array that is "perforated" by just 4 discrete pores (Fig. 3e). The largest, "P1," is formed intramolecularly by the D3-D6 body and encompasses a solvent-accessible area of 778.20 Å² and a maximum cross-sectional diameter of 2 nm. Three additional pores are formed by intermolecular contacts in the lattice. Pores of 2 nm diameter or less would be compatible with the passive passage of small molecules, but not folded proteins.

## Effect of EA1 nanobodies on living bacteria and growth

Given that selected EA1 nanobodies (633, 632, and 643) were able to depolymerize the EA1 S-layer in vitro, we investigated their activity on *B. anthracis* cell morphology and growth. We incubated stationary phase *B. anthracis* strain 7702 cells (when EA1 levels are higher) with an equimolar mixture of the selected depolymerizing nanobodies and evaluated the integrity of the S-layer using TEM. It is important to emphasize that analyzing S-layer integrity via negative-

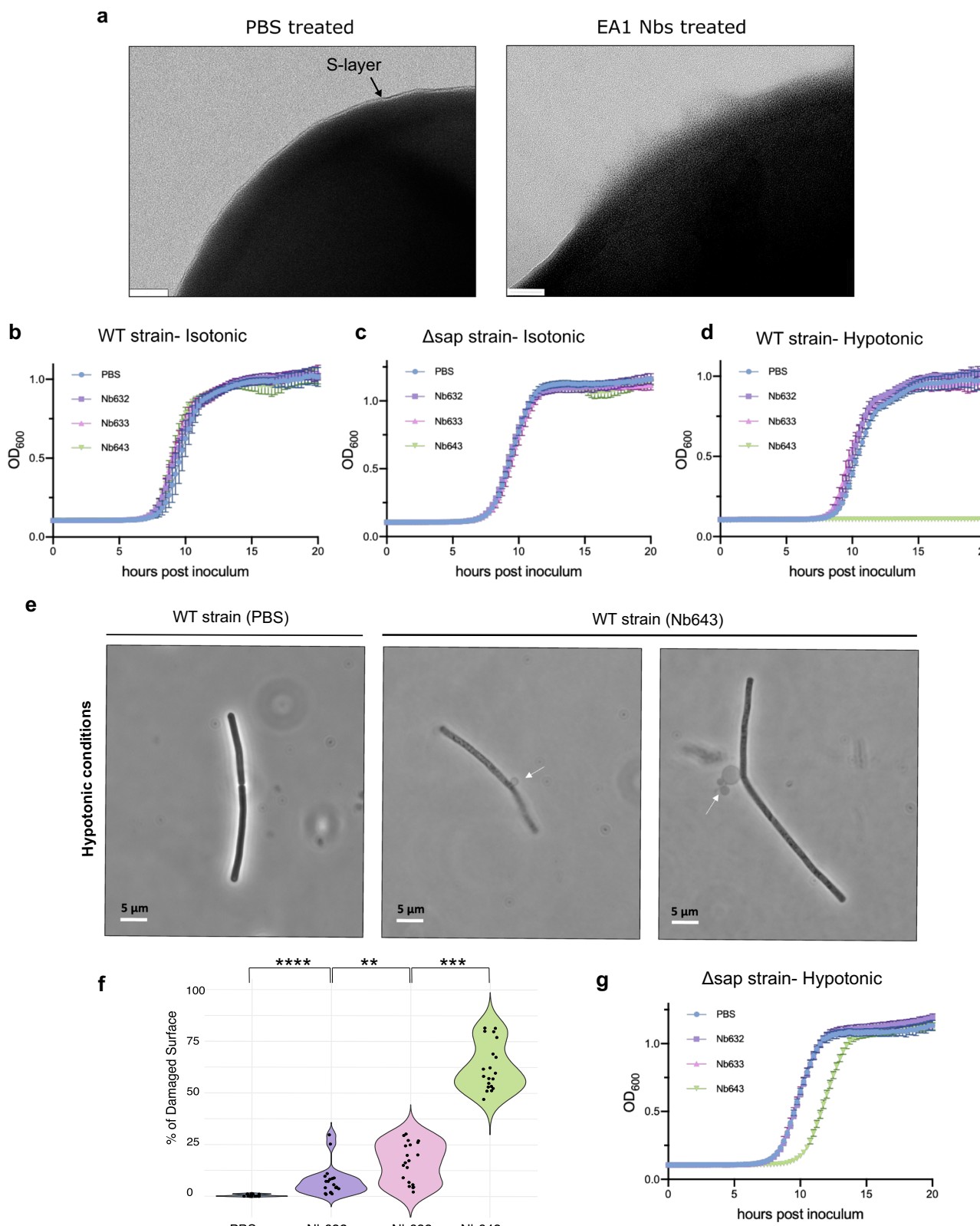

staining TEM may not be fully representative due to potential alterations in the S-layer structure caused by the fixating agents used during cell treatment, and will need to be confirmed in vivo. Despite this limitation, our observations of patches of compromised cell surface are consistent and in agreement with our in vitro findings. Control cells treated with PBS presented an intact and continuous S-layer seen as a smooth, high-contrast demarcation of the cell surface (Fig. 5a and Supplementary Fig. 14). Treatment with the Nbs,

however, resulted in a loss of the continuous S-layer and in the formation of irregular perturbances on the cell surface (Fig. 5a and Supplementary Fig. 14). Similar surface defects were previously seen in exponential phase *B. anthracis* cells treated with Sap depolymerizing Nbs[14]. The acute, Nb-induced loss of a crystalline Sap S-layer was found to result in mechanical defects in the cell envelope and an increased vulnerability of cells to hypoosmotic conditions[5].

**Fig. 5 | Effect of the EA1 nanobodies in vivo. a** TEM images of the surface of *B. anthracis* 7702 cells treated with PBS buffer or 200 μM of EA1 nanobodies mixture (Nb633, Nb632, and Nb643). In PBS-treated cells, the S-layer is shown as a continuous high-contrast monolayer on the bacterium's surface (indicated with an arrow). In EA1 nanobodies-treated cells, the S-layer is compromised indicated by the loss of the ordered monolayer. The scale bar is 40 nm. The experiment was repeated three independent times. **b, c** Growth curves of *B. anthracis* 7702 WT or *Δsap*. Cells were incubated with PBS (control) or EA1 nanobodies (Nb633, Nb632 and Nb643 at 200 μM each) prior to inoculation in BHI media. **d** Growth curves of *B. anthracis* 7702 WT cells treated with PBS (control) or single EA1 nanobodies (Nb633, Nb632 and Nb643 at 200 μM each). Cells were incubated for 40 minutes in deionized H₂O (hypotonic condition) prior to inoculation in BHI. **e** Phase contrast images of *B. anthracis* 7702 cells treated with Nb643 (200 μM) or PBS (control) and

exposed to hypotonic conditions for 40 min. Nanobody-treated cells show cytoplasmic membrane bulges (white arrows) and loss of contrast, indicative of cell lysis. The data shown are representative of experiments made independently in triplicate. **f** Violin plots showing the quantification (in %) of the damaged surface layer of WT cells treated with single Nbs (200 μM) or PBS (control), screened using negative stain TEM, each point represents a single cell. Twenty cells were analyzed for each condition. Statistical analysis by two-sided Mann–Whitney *U*-test, with P values corresponding to ∗∗∗∗P = 1.4509e−14, ∗∗∗P = 4e−10), ∗∗P = 0.021. **g** Growth curves of *B. anthracis Δsap* cells treated with PBS (control) or single EA1 nanobodies (Nb633, Nb632 and Nb643 at 200 μM each). Cells were incubated for 40 minutes in deionized H₂O (hypotonic condition) prior to inoculation in BHI. Growth curve data are sample mean ± s.d., representative of *n* = 3 biologically independent experiments.

We next evaluated if Nb-induced disruption of the EA1 S-layer affected cell growth or viability. Since EA1 is expressed during late exponential growth, we followed the outgrowth of inocula of *B. anthracis* strain 7702 (WT) taken from stationary phase cells transferred into fresh medium with or without prior treatment with EA1 depolymerizing nanobodies (i.e., 200 μM of Nb632, Nb633 or Nb634 in PBS, or just PBS, prior to transfer into fresh BHI). Under these conditions, none of the tested nanobodies resulted in an altered outgrowth of the inocula compared to cells treated with buffer (Fig. 5b). The same was observed for *B. anthracis* strain RBA9 (i.e., *Δsap*), an isogenic mutant lacking *sap*[29], thus ruling out the possibility that a compensatory expression of the Sap S-layer would obscure any growth phenotype of EA1 S-layer disruption (Fig. 5c). In light of recent observations that the Sap S-layer provides mechanical support to the cell envelope of cells in exponential growth[5], we reasoned that EA1 may fulfil a similar function during late exponential and/or stationary phase. We therefore submitted buffer- or Nb-treated stationary phase cells to an osmotic downshock prior to inoculation of fresh BHI media. Under hypotonic conditions, treatment of cells with Nb643, but not Nbs 632 or 633, resulted in a complete loss of cell viability (Fig. 5d). When observed by phase contrast microscopy, Nb643 treated cells that underwent an osmotic downshock showed extensive blebbing and lysis of the cell membrane (Fig. 5e). In contrast, cells treated with buffer control did not show lysis or a loss of viability upon osmotic downshock (Fig. 5e). These combined data indicate that similar to Sap, the acute disruption of the EA1 S-layer compromises the ability of the cell envelope to resist turgor pressure. Although Nbs 632 and 633 depolymerize EA1 S-layers in vitro (Fig. 1d,e), they do not appear to do so efficiently in vivo (Fig. 5f). Whereas treatment with Nb643 resulted in an average 62% damaged surface area, treatment with Nb632 or Nb633 resulted in only 7 and 16% damaged surface area, respectively. Based on the EA1 monomer and lattice structure it emerges that Nb643 binds the hinge of D1-D2, a site that remains accessible in the S-layer lattice, whereas Nb632 (a structure of EA1 in complex with Nb633 is not available) binds a surface area that becomes inaccessible in the assembled S-layer (i.e., D2–D5 contact; Fig. 2a). Likely, the in vitro depolymerization activity of Nb632 stems from binding and lattice destabilization from the edges of S-layer flakes, whereas the continuous nature of S-layer on the cell may preclude efficient binding and/or S-layer destabilization of the Nb.

Remarkably, when cells of the *Δsap* strain were treated with Nb643 and exposed to an osmotic downshift, growth curves only showed a partial loss in viability in the form of an extended lag phase (Fig. 5g). Repeated treatment cycles showed a reproducible delay in the lag phase, showing *Δsap* cells were less prone to osmotic lysis upon loss of the EA1 S-layer, thus suggesting the presence of an osmoadaptive mechanism in the mutant lacking Sap.

## Discussion

In this study, we report the experimentally determined structure of EA1, the S-layer of the human and animal pathogen *B. anthracis*. We provide a high-resolution X-ray structure of the EA1 assembly domain

stabilized by S-layer disrupting Nbs and report the medium-resolution cryoEM reconstruction of the EA1 S-layer. These data allow us to understand the organization of EA1 inside S-layer lattices with atomic detail, and describe the mode of action of bioactive nanobodies. Similarly to many other bacterial and archaeal S-layers, the assembly domain of EA1 represents a beads-on-a-string organization of independent domains with an immunoglobulin-like (Ig-like) fold. These domains organize into a flat, tile-like supertertiary structure that forms the assembly competent unit of the SLPs. Remarkably, this multidomain architecture and organization seems to be recurrent in SLPs of the genus *Bacillus*, despite their very low sequence identity across species[21,14,30]. Available and predicted structures of S-layer proteins in the order *Bacillaceae*, including *Geobacillus stearothermophilus* S-layer proteins SbsA, SbsB, SbsC and SgsG, *Lysinibacillus sphaericus* SbpA and *Bacillus anthracis* Sap and EA1, encompass an N-terminal triple helix or SLH domain responsible for cell surface attachment, followed by an assembly domain of 6–8 Ig-like domains. The X-ray structures of the SLH domains of *B. anthracis* Sap and *Paenibacillus alvei* SpaA in complex with the SCWP moiety have provided atomic insight into the non-covalent anchoring of these S-layer proteins to the cell wall[31,32]. Comparison of the predicted model of the EA1_{SLH} to these SLH structures indicates that the fold, as well as key residues for substrate binding, are conserved, suggesting these proteins have a similar binding mode (Supplementary Fig. 15). Our structural studies of EA1^{AD} show that outside the SLH, also equivalent Ig-like domains in the S-layer assembly domain of these SLPs (i.e., D1–D1, D2–D2, etc.) share a striking structural similarity (Supplementary Figs. 16 and 17), indicating a common ancestry in *Bacillaceae*. Despite the shared domain structure and structural conservation of individual domains, however, these proteins show remarkably divergent quaternary structure and lattice properties. The occurrence of beads-on-a-string Ig-like domains has also been observed in the Csg S-layer protein of the archaeon *Haloferax volcanii*[33] (Supplementary Fig. 18) or the HPI S-layer protein of the diderm bacterium *Deinococcus radiodurans*[34] (Supplementary Fig. 19). However, different to the *Bacillaceae* SLPs, the equivalent Ig-like domains show no apparent structural homology (Supplementary Figs. 18 and 19), suggesting that in these distant genera, the architecture consisting of consecutive Ig-like folds is more likely the result of convergent evolution than a common ancestry. In that respect, the occurrence of a beads-on-a-string Ig-like domain architecture is quite common in otherwise non-homologous surface proteins in bacteria, archaea and eukaryotes, including, amongst many others, Ig-superfamily cell adhesion molecules (CAMs)[35], cadherins[36], and different families of bacterial pili[37]. There is no current indication or evidence, however, that these families would represent homologous structures.

We performed an in-depth structural comparison analysis in the *Bacillaceae* group of the available X-ray structures. Ig-like domains in EA1, Sap and SbsB are individually similar (Supplementary Figs. 16 and 17), with Dali calculated Z-scores ranging from 5.1 to 10.5 and averaged rsmds <8 Å. D6 is the most diverse domain with a very low sequence

identity (11.2%) and low structure similarity (rmsd = 15.88 Å). Despite this structural homology and the common buildup of the SLP subunits, they assemble into S-layers with remarkably different lattice properties and symmetries[19,38–40]. Intramolecular and intermolecular domain contacts in these different S-layer proteins depend on the loop regions of the Ig-like domains. These loop regions are structurally plastic and greatly differ amongst the three proteins, resulting in unique super-tertiary structures of the Sap, EA1 and SbsB monomers, and resulting in non-conserved, heteromorphic lattice contacts. The physiological reasons behind the non-conserved lattices and the change in S-layer proteins from species to another or from one growth state to another are largely unknown. Many S-layers, including non-homologous SLPs outside *Bacillaceae* require calcium for the folding of the monomers and/or assembly of the S-layer. In SbsB four calcium ions bind interfacial loops and condensate the SLP monomer into its assembly-competent conformation by ordering and stabilizing intramolecular domain–domain contacts[21]. Our structural analyses suggest an equivalent process occurs in EA1, where three calcium binding sites are located in loop regions that form intramolecular domain–domain contacts. Binding of calcium was essential for EA1 S-layer formation and likely represents the structuring of these intramolecular domain contacts. We speculate that the bead-on-a-string architecture of consecutive Ig-like domains and their structuring via non-conserved calcium binding loops provides an optimized balance between stability at the level of the individual domains, i.e., sufficiently robust to withstand changing extracellular conditions, and the ability to incorporate plastic surface loops that facilitate the formation of a multitude in lattice types. Accordingly, different S-layers do not involve isomorphous or conserved lattice contacts zones. Moreover, individual intermolecular domain interfaces in the S-layer tend to be limited in surface area and weak in affinity, often encompassing only a few, usually polar, contacts. Still, avidity effects result in robust S-layer lattices that provide mechanical support to the cell envelope. At the level of single protomer–protomer contacts, weak intermolecular interfaces may be required for the remodeling of the S-layer during cell extension and division. These weak intermolecular protomer contacts also explain how binding of small affinity reagents such as Nbs, or the chelation of the calcium ions, are able to destabilize and disintegrate S-layers. In this work, we show that binding of a Nb to D2 results in repositioning it away from its assembly-competent conformation, where it would interact with D5 and D6 in neighboring protomers. Also in the Sap S-layer, binding of a Nb in the D1-D2 hinge regions is sufficient to induce S-layer disintegration[14]. These features of a rapid, induced depolymerization result in a loss of cell wall supporting activity when the S-layer lattice transitions from a crystalline to an amorphous state and provides interesting therapeutic opportunities in pathogenic S-layer positive bacterial species.

## Methods

### Cloning for recombinant protein production in *E. coli*

Several variants of *Bacillus anthracis* EA1 (UniprotKB: P94217) were produced in this study with an N-terminal 6xHis tag. The assembly domain (EA1$_{AD}$) encompasses residues from 214–862 and a full-length (EA1$_{FL}$) version with amino acids 31–862. Synthetic codon-optimized EA1$_{AD}$ was cloned into a pET300 leading to the pAFSLP10 plasmid as previously described[14]. The expression plasmid A93 containing EA1$_{FL}$ was assembled by amplifying EA1$_{FL}$ from the *Bacillus anthracis* (strain 7702) genome using pair of primers 368 and 369. The PCR fragment was cloned into a linearized pASK-IBA3plus vector (using primers 321 and 322) by Gibson assembly and transformed into chimio-competent DH5α *E. coli* (New England BioLabs). EA1 mutants were derived from the A93 plasmid using direct point mutagenesis. Double mutant N614W-A484Q was obtained using the primers from 456 to 461, leading to the plasmid A129. Whereas a combination of cysteine mutants was obtained using the primers from 462 to 469, leading to A126

(N703C/N340C), A127 (N703C/D343) and A128 (N703C/K372C). Mutant EA1$_{\Delta D6}$ (residues 31–718) was cloned into the pASK-IBA3plus vector by PCR amplification (template A93) using the pair of primers 368 and 453, followed by Gibson assembly. Finally, mutant EA1$_{\Delta D1}$ (residues 110–862) was cloned into the pASK-IBA3plus vector by PCR amplification (template A93) using the pair of primers 446 and 455, followed by Gibson assembly. For cloning purposes, all the strains were grown in LB at 37 °C and supplemented with 100 μg/ml of Ampicillin when required. All plasmids were sequence-verified (Eurofins) using primers 370, 371, 372, 305 and 306. All plasmids and primers used in this study are listed in Supplementary Table 2.

### Protein expression and purification

EA1 and all EA1 mutants were expressed in *E. coli* BL21 (DE3) grown in Terrific Broth (TB) supplemented with 100 μg/ml of Ampicillin at 37 °C and induced with 1 mM IPTG or 200 μg/L anhydrotetracycline when OD$_{600}$ reached 0.6. At this point, the temperature was set at 25 °C and cells were left to express overnight the desired protein. Next day, cells were harvested by centrifugation and pellets were kept at −20 °C. Frozen pellets from EA1$_{FL}$ and derivative polymerizing EA1 (EA1$_{AD}$, EA1$_{N703C/N340C}$, EA1$_{N703C/D343}$ and EA1$_{N703C/K372C}$) were resuspended in 100 ml of lysis buffer (50 mM Hepes pH 8, 300 mM NaCl, 1 mM MgCl2, DNase, lysozyme and EDTA-free protease inhibitor cocktails (ROCHE)) at 4 °C and lysed by sonication. The lysate was centrifuged for 60 min at 30,000 × *g* at 4 °C. The EA1-containing pellet was resuspended with a homogenizer in Buffer A-Urea (300 mM NaCl, 10 mM Hepes pH 8, 10 mM Imidazole and 8 M urea) and the homogenate was centrifuged for 60 min at 30,000 × *g* at 4 °C. The cleared supernatant containing soluble 6xHis-tagged EA1 was mixed with 5 ml of 40 IDA$^{high}$ agarose IMAC beads charged with Ni2 + (Bio-Works) and pre-equilibrated with Buffer A-Urea and incubated at room temperature for 30 min while rotating. The mixture was centrifuged at 4000 × *g* for 10 min at 4 °C and the pellet was extensively washed with ice-cold Buffer A (30 mM NaCl, 10 mM Hepes pH8, 10 mM Imidazole) in order to remove the urea. The protein was eluted using 100% ice-cold Buffer B (300 mM NaCl, 10 mM Hepes pH8, 1 M Imidazole), filtered with a 0.2 μm filter (Acrodisc LC 13 mm, syringe filter, Life Science) and applied to a Superdex 200 16/60 size-exclusion column (GE Life Sciences) that was equilibrated with 10 mM Hepes pH 8 and 100 mM NaCl at 4 °C. To avoid the formation of the EA1 S-layer after purification, fractions corresponding to EA1 were pooled and immediately flash-frozen in liquid nitrogen and kept at −80 °C. Alternatively, the concentration of pooled fraction was measured and depolymerizing nanobodies were added to a molar excess of 1.2 ratio. When EA1 S-layers were wanted, pooled fractions were concentrated and maintained at room temperature until further use.

Non-polymerizing EA1 mutants (EA1$_{N614W-A484Q}$ and EA1$_{\Delta D6}$) were purified without denaturing agents. Briefly, frozen bacterial pellets were resuspended in 100 ml of lysis buffer and lysed by sonication at 4 °C. The lysate was centrifuged for 30 min at 30,000 × *g* at 4 °C. The cleared supernatant containing EA1 mutant was loaded onto a 5 ml Ni-NTA affinity chromatography column (HisTrap FF crude, GE Healthcare). The column was washed with 5 column volumes with Buffer A and finally, His-tagged proteins were eluted with a linear gradient of Buffer B. Eluted protein was concentrated and loaded onto a Superdex 200 16/60 size-exclusion column (GE Life Sciences) that was equilibrated with SEC buffer (10 mM Hepes pH 8 and 100 mM NaCl) at 4 °C. The peak corresponding to the protein was concentrated, flash-frozen in small aliquots in liquid nitrogen and stored at −80 °C. After purification, the samples were run on a sodium dodecyl sulfate–polyacrylamide gel electrophoresis (SDS-PAGE) to evaluate their purity. The uncropped SDS-PAGE gels are shown in Supplementary Fig. 20.

### EA1$_{AD}$ nanobodies isolation, production and purification

As described in ref. [14], a single individual of llama (*lama glama*) was immunized weekly with a 6 subcutaneous injection of adjuvant (Gerbu

LQ, GERBU biotechnik) emulsified EA1$_{AD}$ (0.5 mg per injection), each administrated within 15 h of purification and kept at 4 °C to ensure the maximal amount of monomeric EA1$_{AD}$. Four days after the final injection, total RNA was extracted from peripheral blood mononuclear cells in a 50 ml blood sample as described in ref. [41]. Llama immunization was performed in accordance with institutional guidelines, following an experimental protocol reviewed and approved by the Vrije Universiteit Brussel Ethical Committee for Animal Experiments (project number 16-601-3). Starting from total RNA, cDNA was synthesized and the Nb repertoire was amplified and cloned as described previously[42], except that phagemid pMESy4 was used as the display vector, enabling the expression of C-terminal 6-His-EPEA tagged Nbs. The resulting phage library consisted of approximately $4.6 \times 10^9$ independent clones containing an insert that corresponded to the size of a nanobody. To identify EA1$_{AD}$-specific binders, 1 µg of the monomeric EA1AD antigen was solid-phase immobilized in sodium bicarbonate buffer pH 8.2 in a 96-well Maxisorp plate (Nunc). Microwells were subsequently blocked using PBS containing 2% skimmed milk powder before incubation with the phage library. Unspecific phages were removed by extensive washing with PBS containing 0.05% Tween-20, and bound phages were eluted after trypsin treatment. Two such selection rounds were performed and 94 single clones were randomly picked from outputs of the first and second rounds. *E. coli* WK6 were transformed with purified plasmids from selected clones that were grown on LB medium and induced using IPTG for periplasmic expression of monoclonal Nbs as described elsewhere[43]. The specificity of the Nbs was verified using ELISA by coating 1 µg of the monomeric EA1$_{AD}$ in Maxisorp plates. Bound Nbs were detected with the EPEA tag using a CaptureSelect Biotin anti-C tag conjugate (Life technologies) mixed with alkaline phosphatase (Promega) for detection. EA1$_{AD}$-specific Nbs were sequenced, resulting in a library of 16 unique monoclonal Nbs belonging to seven different families. Sequences for Nbs used in this study are found in Supplementary Fig 4. Selected Nbs were expressed and purified as described previously with the exception that size-exclusion chromatography was performed as the final purification step for each Nb with SEC buffer. Purified Nbs were buffer exchanged for PBS to perform bacterial experiments.

## Circular dichroism, polymerization assays, and DLS

The secondary structure of EA1 after purification with 8 M urea was determined using circular dichroism (CD) on a Circular dichroism spectrometer MOS500 (BioLogic). Three individual scans were averaged to obtain a final spectrum. Measurements were made at 25 °C with a 1 mm Quartz suprasil cuvette (Hellma Cells 110-Qs 1 mm 110-1-40). To carry out the experiment, 200 µl of EA1 at 1 mg/ml were used in a buffer containing 150 mM NaCl and 10 mM Hepes pH8. The same buffer was used to measure the blank.

To follow the depolymerization effect of the nanobodies on the EA1 S-layer, the absorbance at 400 nm was measured over time. The experiment was carried out in a 96-well plate (Cellstar; Greomer bioone) and the absorbance was measured using the in the Cytation One (Biotek) with an emission and excitation light of 400 nm. Measurements were taken for 10 min every 15 s with low shaking. To perform the experiment, EA1 at 2 mg/ml (25 µM) was let to polymerize at room temperature for 12 h. The next day, 70 µl EA1 were mixed with 30 µl of nanobody to have a final concentration of 18 µM and 350 µM respectively (final ratio 1:20) in 100 µl final volume. Measurements were started immediately. Both EA1 and nanobodies are found in SEC buffer and the experiment was repeated at least three times.

Finally, dynamic light scattering (DLS) was used to follow the presence or absence of S-layers. Before DLS experiments, protein samples were centrifuged at $20,000 \times g$ for 15 min, EA1 (and mutants) were in SEC buffer and used at a final concentration of 2 mg/ml, and the experiments were collected at 25 °C in 4 µl cyclic olefin copolymer disposable cuvettes at an angle of 90° using a Dynapro NanoStar DLS

machine (Wyatt Technology). The Dynamics software (version 7.1.9.3) was used to schedule data acquisition and data analysis. For each sample, 20 measurements of 10 s were averaged and this operation was repeated 20 times for each condition. The plots represent the distribution of the intensity over the diameter of the particles.

## Bio-layer interferometry

To measure the interaction of the Nbs with the non-polemizing EA1$_{\Delta D6}$, label-free bio-layer interferometry experiments were performed on the BLI Octet instrument (ForteBio, Inc., USA) at room temperature. Both EA1$_{\Delta D6}$ and Nbs (643, 633 and 632) were buffer exchanged with BLI buffer (150 mM NaCl, 10 mM Hepes pH 7.5 and 0,03% DDM). To biotinylate EA1$_{\Delta D6}$ in order to bind onto the streptavidin Octet SA biosensors (Sartorius), 100 µM of EA1$_{\Delta D6}$ were incubated with 500 µM of EZ-Link NHS-PEG4-Biotin (ThermoFisher Scientific) during 1 h at room temperature. To remove the excess of EZ-Link NHS-PEG4-Biotin the protein was desalted in BLI buffer using a Zeba Spin Desalting Column 7 K MWCO Biotin (ThermoFisher Scientific) following the manufacturer's instructions. Biotinylated EA1$_{\Delta D6}$ at 3 µg/ml was used to load the sensors. Nb 643, 633, and 632 were serial diluted from 50 nM to 0,78 nM. After loading the sensor with EA1$_{\Delta D6}$ for 30 s, the sensor tip was placed for 60 seconds in BLI buffer to remove any loosely bound protein. The next step was the association with the Nb for 15 min followed by a dissociation step of 15 min. Sensors were regenerated by soaking them for 15 seconds in 75 mM of orthophosphoric acid. Data were automatically collected by the BLI system and analyzed with GraphPad Prism 8. Assays were performed at least by duplicate, and to obtain the apparent $K_d$ the binding signals at the end of the association step were plotted against the protein concentrations using a bivalent nonlinear regression fitting.

## Western blot

To prepare the sample, 30 µg of each recombinant cysteine mutant (EA1$_{N703C/N340C}$, EA1$_{N703C/D343}$, and EA1$_{N703C/K372C}$) were boiled with and without 1 mM of 2-mercaptoethanol. Samples were analyzed by SDS-PAGE, electro-transferred onto a 0.2 µm Nitrocellulose membrane and blocked with 5% (w/v) skimmed milk for 1 h. The membrane was incubated for 1 h at room temperature with Mouse anti Histidine Tag:Alkaline phosphatase-conjugated (BioRad MCA1396A). After washing three times in TBS-Tween buffer (Tris-HCl pH 8 10 mM; NaCl 150 mM; Tween20 0.05% (vol/vol) for 5 min each wash. The membrane was revealed by incubating for 10 min with NBT/BCIP (Roche) and imaged using the GelDoc imager (Biorad). Prestained PageRuler (Thermo Fisher Scientific) was used as a molecular ladder. The uncropped blot is shown in Supplementary Fig. 20.

## Crystallization, structure determination and analysis

Crystallization screens were performed using freshly purified EA1$_{AD}$ in combination with several nanobodies at 1.2-fold molar excess, and the mixture was concentrated using an AMICON 10 kDa MWCO to 45 mg/ml. Optimal crystals of EA1$_{AD}$ in complex with Nb643 and Nb632 were obtained using the sitting drop vapor diffusion method and a Mosquito nanoliter-dispensing robot at room temperature (TTP Labtech, Melbourn, UK). Thin plate-shaped crystals appeared after 3 months in a condition containing 2 M ammonium sulfate and 0.1 M sodium acetate pH 5.0. The crystallization buffer was supplemented with 10% glycerol, and crystals were mounted in nylon loops and flash-cooled in liquid nitrogen. X-ray diffraction data were collected at 100 K using the Beamline Proxima 2 (wavelength = 0. 9801 Å) at the Soleil synchrotron (Gif-sur-Yvette, France). Data was shown to be anisotropic and we processed it using AutoProc and Staraniso[44] at 1.81 Å in P1 with unit-cell dimensions of $a = 72.927$, $b = 74.297$, $c = 87.649$, $\alpha = 107.853$, $\beta = 101.138$, and $\gamma = 112.42$. The crystal structure was determined by molecular replacement using phaser from the phenix suite[45,46] and using as search models the single domains of EA1

and the two nanobodies predicted by AlphaFold2[25]. The structure was refined through iterative cycles of manual model building with COOT[47] and reciprocal space refinement with phenix.refine[48] and Buster[49] to $R$ values of $R_{work}/R_{free}$ of 0.19/0.23. The crystallographic statistics are shown in Supplementary Table 1. Structural Figures were generated with ChimeraX[50]. Atomic coordinates and structure factors have been deposited in the protein data bank (PDB) under the accession code 8OPR.

### Negative-stain transmission electron microscopy (TEM)
Purified EA1 was concentrated at 2 mg/ml and let to polymerize for 24 h before preparing the TEM grids. For visualization of EA1 S-layers by negative stain TEM, carbon-coated copper grids with 400-hole mesh (Electron Microscopy Science) were glow discharged (ELMO; Agar Scientific) with a plasma current of 5 mA at vacuum for 60 s. Freshly glow-discharged grids were used immediately by applying 4 μl of sample and allowing binding to the support film for 1 min after which the excess liquid was blotted away with Whatman grade 2 filter paper. The grids were then washed three times using three 15 μl drops of ddH$_2$O followed by blotting of excess liquid. The washed grids were held in 15 μl drops of 2% uranyl acetate three times for, respectively, 10 s, 2 s, and 1 min duration, with a blotting step in between each drop. Finally, the uranyl acetate coated grids were fully blotted. The grids were then imaged using a 120 kV JEOL 1400 microscope equipped with LaB6 filament and TVIPS F416 CCD camera.

### Preparation of Cryo-EM grids, data collection and image processing
Quantifoil ® holey Cu 300 mesh grids with 2 μm holes and 1 μm spacing were first glow discharged in a vacuum using plasma current of 5.5 mA for 1 minute (ELMO, Agar Scientific). For cryo-plunging, a Gatan CP3 cryo-plunger station was used. To prepare the grids, 3 μl of polymerized EA1$_{FL}$ S-layer at 2 mg/ml was applied onto the grid at room temperature and 90% humidity. After 90 s, the liquid was machine-blotted with Whatman grade 2 filter paper for 4.5 s from both sides and plunged frozen into liquid ethane. Grids were stored in liquid nitrogen until data collection. One dataset was collected on EA1$_{FL}$ S-layers. High-resolution cryo-EM 2D micrographs movies were recorded on a JEOL Cryoarm 300 microscope automated with an energy filter and a K3 direct electron detector run in counting mode at the BECM facility (Brussels, Belgium). The dataset was recorded on a K3 detector, at a pixel size of 0.782 Å/pxl, and exposure of 64.66 e-/Å² taken over 61 frames/image at 60 K magnification. A total of 7095 movies were recorded. Briefly, images were imported to cryoSPARC[51] where they were motion-corrected using Patch Motion Correction and defocus values were determined using Patch CTF. Particles were picked using blob picker and extracted with a box size of 400 × 400 pixels. Several rounds of 2D classification were needed in order to clean selected particles providing a set of 309.149 high-quality particles. Several Ab-initio model were generated and the model with clear lattice features was selected (composed of 183.113 particles) and subjected to homogenous refinement and non-uniform refinement yielding to a map of 6.61 Å resolution as estimated based on the gold-standard Fourier shell correlation using the 3DFSC server[52]. No symmetry was imposed during data processing. Coordinates for the model of the EA1$^{AD}$ S-layer lattice is available through Mendeley at DOI: 10.17632/wf93wvy2vz1. are available upon request.

### Effect of EA1 nanobodies on *B. anthracis* cell surface and growth
For negative stain TEM images of the bacteria cell surface, *B. anthracis* strain 7702 cells were grown overnight in BHI medium at 37 °C. 50 μl of cells were centrifuged at 5000 × *g* for 5 min and the pellet was resuspended in 0.5 ml of PBS supplemented or not with a mixture of Nb632, Nb633 and Nb643 (200 μM final concentration each). Cells were incubated for 20 min at room temperature, washed with PBS and fixed for 1 h with 2% PFA. Cells were washed with PBS and stained with 1% of uranyl acetate as described above. Negatively stained cells were then imaged using an in-house 120 kV JEM 1400 (JEOL) microscope equipped with a LaB6 filament and a CMOS camera (TVIPS TemCam F-416). For growth curves, *B. anthracis* strain 7702 and Δ*sap* cells were taken from an overnight culture (OD$_{600}$ > 2; maximum expression of EA1), harvested by centrifugation (3 min at 18,000 × *g*) and resuspended in PBS to a final OD$_{600}$ of 0.2. Cells were incubated with PBS (control) or 200 μM final concentration of each nanobody in a final concentration of cells of OD$_{600}$ 0.025. The mixture was incubated at room temperature for 40 min. For the isotonic treatment, 200 μl of the above mixture (final OD$_{600}$ 0.00025) were incubated in PBS for 45 min at room temperature whereas cells that were subjected to a hypotonic shock were placed in water. Finally, cells were exposed to a brief procedure of increased shear by pipetting 5 times, before inoculation in a 96-well plate (Falcon) with fresh BHI media to a final OD$_{600}$ 0.0000125 in order to start the growth curves measurements. Plates were placed in the Cytation One (BioTek) and growth conditions were set at 37 °C under double orbital shaking. OD$_{600}$ measurements were taken every 15 min. Data was analyzed and plotted using Prism8.

### Phase contrast microscopy
*B. anthracis* strain 7702 cells exposed to PBS or to 200 μM of Nb643 treatment for 45 min prior to hypotonic conditions (deionized water for 45 min) were placed on a 2% agarose pad prepared with PBS. Cells were visualized using the phase contrast mode on a Leica DMi8 inverted microscope (Leica) with ×100/1.32 oil objective (Leica).

### Reporting summary
Further information on research design is available in the Nature Portfolio Reporting Summary linked to this article.

## Data availability
Data that support this study are available upon request. The atomic coordinates have been deposited in the Protein Data Bank (PDB) under accession code 8OPR (crystal structure of EA1$^{AD}$ in complex with nanobodies 632 and 643). The model for the EA1 S-layer lattice is available through Mendeley at https://data.mendeley.com/datasets/wf93wvy2vz/1. The cryo-EM density map is available from the authors upon request. Data and materials are available from the corresponding authors upon request. Requests for *Bacillus anthracis* strains should be addressed to the primary source, as cited in the manuscript. Source data are provided with this paper.

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

## Acknowledgements

This work was partially supported by VIB and FWO Flanders through project grant number G065220N granted to A.F. and H.R. A.F. is supported by a post-doctoral fellowship from FWO (1253121N) and a BOF-

ZAP professorship from VUB; A.S. was supported by FWO grant G065220N and an EMBO post-doctoral fellowship (ALTF-709-2021). We thank INSTRUCT-ERIC and the Research Foundation - Flanders (FWO) for their support to the Nanobody discovery as well as Nele Buys for the technical assistance during Nanobody discovery. We thank Dr. Mike Sleutel for his helpful insights into cryo-EM data analysis and discussions. We thank BECM and Dr. Marcus Fislage for his assistance during cryo-EM data collection. We thank the staff of the synchrotron SOLEIL for assistance and support in using beamlines PX1 and PX2.

## Author contributions

A.F and A.S. performed cloning and protein purification; A.S. performed biophysics experiments and analyzed the data; A.F. performed nanobody identification; E.P. and J.S. supervised llama immunization and identification of nanobodies; W.J. assisted in protein production; A.S. performed structural studies supervised by H.R.; TEM and cryo-EM experiments were performed by A.S.; experiments with life bacteria were performed by A.S. and A.F.; A.F., A.S. and H.R. wrote the manuscript; all authors edited the final version of the manuscript.

## Competing interests

H.R. and A.F. are mentioned as inventors on a patent describing the use of S-layer destabilizing antibodies as anti-infectious agents. Publication number: 20200255501; Publication Date: Aug 13, 2020; Patent Grant number: 11377484; Inventors: Han Remaut (Roosbeek), Antonella Fioravanti (Brussel); Application Number: 16/651,713. The other authors declare no competing interests.
