## [Peer Review File · Nature Communications]

Structure and function of the EA1 surface layer of *Bacillus anthracis*Reviewers' Comments:

Reviewer #1:

Remarks to the Author:

In this study, Sogues and co-authors described the structure of the surface layer protein EA1 of *Bacillus anthracis*. They were able to report the structure of the EA1 assembly domain, something that was assayed for years, by using nanobodies that inhibit EA1 self-assembly and depolymerize existing EA1 S-layer lattices. This is an important knowledge for the field. The manuscript is well written and well organized.

My only major comment concerns the last paragraph of the result section: Effect of EA1 nanobodies on living bacteria and growth. In this paragraph, the authors observed that *B. anthracis* cells did not show a reduced growth in the presence of EA1 depolymerizing Nbs. They hypothesized that the loss of EA1 S-layer can be compensated by SAP expression. This can easily be tested by performing the same experiment with the Δ sap strain (strains that was used by the authors in a previous publication (Fioravanti, Mathelie-Guinlet, Dufrene, Remaut, 2022)). It would be appreciated to include this in the manuscript. Also, what happen when EA1 Nbs are added to growing cells when they enter in stationary phase (when EA1 is already on the cell surface)? A growth curve experiment can be performed as the author did (growth in BHI in microplate) but instead of adding the nanobodies at the beginning of the growth experiment, Nbs can be added at the entry of stationary growth phase.

Minor correction:

Line 85 « (SLH) that anchor the protein to the cell surface through binding of the ketal pyruvylated N-acetylmannosamine unit in the peptidoglycan via non-covalent interactions” Please replace peptidoglycan by Secondary Cell Wall Polysaccharide (SCWP)

End of line 147, a bracket is missing

L194 and l196, please replace fig 2g by fig 2f

Line 293, replace levers by levels

Line 327, correct EA by EA1

Line 338, correct heteropmorphic by heteromorphic

Line 600, a word is missing after 20, is it 20 min?

Reviewer #2:

Remarks to the Author:

Review for Structure of the EA1 surface layer of *Bacillus anthracis*, Sogues et al

The authors report the S-layer structure of the EA1 S-layer from *Bacillus anthracis* using X-ray crystallography and docking into a cryo-EM map. The authors further show that nanobodies can disrupt the cellular S-layer. I find the results interesting and I recommend publication if the authors could respond reasonably to my comments below -

Major comments

- I am having a major difficulty in seeing the disruption of the S-layer in Figure 5a. This could be due to the usual problems with staining. Have the authors performed cryo-EM? If cryo-EM is not possible, could the authors please show a gallery of images for the control and treatment?
 - o I would recommend randomising the images and asking an unbiased person to quantify and then report statistics.
- Structure of EA1 containing immunoglobulins is fascinating. Could the authors please compare the immunoglobulins to those reported from archaea (PMID: 34818541) and diderm bacteria (PMID: 37043530)?

- Line 116 – how similar is the SLH domain to previously solved structures, e.g Paenibacillus alvei? Please show alphafold's prediction for the SLH domain (PMID: 30087354).
- Line 220 – could you show a map vs model FSC for the D3-D6 fitted into the cryoEM map? How close is it to the reported resolution?
- Lines 248-249 – the mutants suggest that the X-ray structure lattice is a “true contact” but it does serve as a direct proof. Also, please explain briefly in the text what are these mutants ($\Delta D6$ and N614W-A484Q) since this is the first time they are mentioned in the text.
- What happens when the Sap Nbs were mixed with EA1 Nbs? Cell growth should be decreased as the expression of either S-layer proteins cannot compensate for depolymerization.
- Has any Nbs mixture performed better than others at preventing EA1 polymerisation or were single Nbs and mixtures equally potent?
- Line 391 – why was the C-terminal domain truncated in the $\Delta D1$ mutant? It is actually a $\Delta D1 \Delta D6$ mutant. Please correct this in the text.
- Line 541 – please mention that anisotropic correction was performed to obtain the X-ray crystal structure. If this is not the case, why was Staraniso used?

Minor comments

General – doi links of several papers is incorrect (Couture-Tosi, Domanska, Doran, etc), please correct these.

- Line 171 – please add a few words on SbsB for non-specialists.
- Were the authors able to identify where Nb633 binds EA1?
- Line 228 – Supplementary figure 11 mentions the cysteine mutants, however they are presented later. I would remove any mention until the relevant part of the paper.
- Line 111 – how does the predicted alphafold EA1AD compare with the experimentally determined structure? Please add a superposition of both to supplementary figure 1.
- Line 96 – also cite (von Kügelgen et al., 2020).
- Line 119 – EA1FL (I assume FL = full-length) should be abbreviated.
- Line 120,130 – does recombinant EA1 refer to the EA1FL or EA1AD or both? Please clarify.
- Line 124 – please correct “materials and methods” to “methods”.
- Line 147 – missing) after Nb11.
- Lines 231-232 – add abbreviation to BR, BL, R in the text as well.
- Line 257 – typo, should read SC not SD.
- Line 276 – does CC refer to map vs model? Please clarify.

- Line 279 – DLS, abbreviate.
- Lines 299-365 – should be Sap not SAP.
- Line 338 – typo, should be heteromorphiC.
- Line 555 – was calcium added to the purified EA1 to facilitate polymerization?

Figures

- Figure 3 and Supplementary figure 9 – please indicate where the interacting loops with the bound calcium are present in the lattice. Do they make any connections to the neighbouring subunits? In Supplementary figure 9 it might be better to move/make transparent the 74.29 Å label to better visualize the contacts made by D6 with its surrounding subunits.
- Figure 4 – please show the MC atoms that are involved in H-bond formation in all interfaces presented.
- Supplementary figure 1 – could the authors show the alphafold prediction for Sap as well? How does the confidence score and prediction look like for Sap?
- Supplementary figure 2 – in a) please correct to “(G and S respectively as indicated on the figure)”.
- Supplementary figure 4 – CDR should be abbreviated and mentioned in the text.
- Supplementary figure 7 – b) the superposition is not very informative, perhaps add transparency to each domain to visualize the similarities better. d) It would be interesting to superpose EA1 and Sap to view their overall arrangement.
- Supplementary figure 11 – how do the Cys mutants’ lattices compare with the AD lattice? Perhaps adding Fourier transform of the three double mutant alongside AD would be useful.

REVIEWER COMMENTS

Reviewer #1 (Remarks to the Author):

In this study, Sogues and co-authors described the structure of the surface layer protein EA1 of *Bacillus anthracis*. They were able to report the structure of the EA1 assembly domain, something that was assayed for years, by using nanobodies that inhibit EA1 self-assembly and depolymerize existing EA1 S-layer lattices. This is an important knowledge for the field. The manuscript is well written and well organized.

Author response:

We would like to thank Reviewer #1 for this positive assessment of our work and its contribution to the field.

My only major comment concerns the last paragraph of the result section: Effect of EA1 nanobodies on living bacteria and growth. In this paragraph, the authors observed that *B. anthracis* cells did not show a reduced growth in the presence of EA1 depolymerizing Nbs. They hypothesized that the loss of EA1 S-layer can be compensated by SAP expression. This can easily be tested by performing the same experiment with the Δsap strain (strains that was used by the authors in a previous publication (Fioravanti, Mathelie-Guinlet, Dufrêne, Remaut, 2022)). It would be appreciated to include this in the manuscript. Also, what happen when EA1 Nbs are added to growing cells when they enter in stationary phase (when EA1 is already on the cell surface)? A growth curve experiment can be performed as the author did (growth in BHI in microplate) but instead of adding the nanobodies at the beginning of the growth experiment, Nbs can be added at the entry of stationary growth phase.

Author response:

As suggested by Reviewer #1, we now more elaborately investigate the possible biological activity of EA1 depolymerising Nbs. The growth curve shown in Figure 5 of the original manuscript was done using an overnight, stationary phase culture as inoculum, and therefore representing an inoculum with maximum presence of EA1 when starting the growth curve. However, when resuming growth, the EA1 S-layer is expected to be readily replaced by Sap. We now include the suggested experiment using the Δsap strain, but find no growth delay or defects when using EA1 nanobodies (Figure 5c), thus ruling out the hypothesis that a lack of phenotype from the EA1 nanobodies stems from a compensatory expression of the Sap S-layer. In the Fioravanti et al. 2022 study we identified the Sap S-layer as a mechanical support to the cell envelope, and showed that acute loss of a crystalline Sap S-layer by Nb treatment resulted in cell lysis during hypo-osmotic stress. Therefore, we decided to test the activity of the EA1 nanobodies by exposing the treated cells to an osmotic downshift prior to inoculation in BHI medium. This showed that when exposed to hypotonic conditions, the WT strain fully lost viability when treated with nanobody Nb643 (Figure 5.d). When inspected by microscopy, these cells showed membrane blebbing and cell lysis, indicating that in absence of a crystalline EA1 S-layer the cell envelope has a lowered ability to provide mechanical resistance to the cell turgor.

Remarkably, when cells of the Δsap strain were treated with Nb643 and exposed to an osmotic downshift, growth curves only showed a partial loss in viability in the form of an

extended lag phase (Figure 5.e). Repeated treatment cycles showed a reproducible delay in lag phase, and microscopic inspection showed the Δsap cells were less prone to osmotic lysis upon loss of the EA1 S-layer, thus suggesting the presence of an osmoadaptive mechanism in the mutant lacking Sap.

These new results are in line with observations made in Fioravanti, Mathelie-Guinlet, Dufrêne, Remaut, 2022, and show that an acute loss of a crystalline S-layer results in a cell envelope with reduced mechanical resistance to low osmolality conditions. An effect that can be readily compensated in cells that chronically lack an S-layer, however, by a hitherto unknown mechanism.

Minor correction:

Line 85 « (SLH) that anchor the protein to the cell surface through binding of the ketal pyruvylated N-acetylmannosamine unit in the peptidoglycan via non-covalent interactions”

Please replace peptidoglycan by Secondary Cell Wall Polysaccharide (SCWP)

End of line 147, a bracket is missing

L194 and l196, please replace fig 2g by fig 2f

Line 293, replace levers by levels

Line 327, correct EA by EA1

Line 338, correct heteropmorphic by heteromorphic

Line 600, a word is missing after 20, is it 20 min?

Author response:

We thank the reviewer for pointing these out, and included all the minor corrections in the revised text.

Reviewer #2 (Remarks to the Author):

Review for Structure of the EA1 surface layer of Bacillus anthracis, Sogues et al

The authors report the S-layer structure of the EA1 S-layer from Bacillus anthracis using X-ray crystallography and docking into a cryo-EM map. The authors further show that nanobodies can disrupt the cellular S-layer. I find the results interesting and I recommend publication if the authors could respond reasonably to my comments below -

Author response:

We would like to thank the reviewer for his/her input and positive comments on the manuscript. The reviewers' specific comments are addressed in full below.

Major comments

- I am having a major difficulty in seeing the disruption of the S-layer in Figure 5a. This could be due to the usual problems with staining. Have the authors performed cryo-EM? If cryo-EM is not possible, could the authors please show a gallery of images for the control and treatment? o I would recommend randomising the images and asking an unbiased person to

quantify and then report statistics.

Author response:

In our hands, the thick cell wall found in Bacilli prevents the use of cryoEM. Whereas diderm bacteria tend to show a good electron transparency as full cells, this is not the case for *Bacillus cereus*. To address the reviewers' concern, we collected more TEM images and included a gallery of images for the control (PBS) and the nanobodies-treated cells at different magnifications. The gallery of images is now supplementary Figure 14. In addition, more images using the same methodology are available in the following publications: PMID: 36714836 and PMID: 31308522.

- Structure of EA1 containing immunoglobulins is fascinating. Could the authors please compare the immunoglobulins to those reported from archaea (PMID: 34818541) and diderm bacteria (PMID: 37043530)?

Author response:

It is striking that several S-layer proteins contain Ig-like domains. We expanded Supplementary Figure 16 and 17 by adding an extra panel (c), which presents a dendrogram of the Z-score (structure similarity) between the domains of EA1 compared to Sap and SbsB. Our analysis revealed that the respective single domains from *Bacillus* and *Geobacillus* SLPs Sap, SbsB and EA1 exhibited structural homology, providing a strong indication of a common ancestry. However, when we extended our analysis and included Ig domains from *Haloferax volcanii* Csg S-layer (Supplementary Figure 18) and *Deinococcus radiodurans* HPI (Supplementary Figure 19), we observed that the same domains did not cluster together. This finding suggests that these proteins with Ig-like domains likely evolved independently rather than sharing a common ancestor. We note that Ig-like folds are quite small and relatively simple, yet highly stable folding units, that have convergently emerged in many unrelated proteins, particularly in extracellular proteins.

We have now included a few lines in the main text (discussion) to report this structural comparison.

- Line 116 – how similar is the SLH domain to previously solved structures, e.g *Paenibacillus alvei*? Please show alphafold's prediction for the SLH domain (PMID: 30087354).

Author response:

We have included a sequence alignment containing the SLH domains of EA1, Sap and SpaA from *Paenibacillus alvei*, and observed that conserved residues crucial for SCWP binding are shared among these three species. In addition, we have also included a high-accuracy prediction (pLDDT > 90) of EA1 SLH domains (supplementary Figure 15) and performed a structural alignment with the SLH domains from Sap and SpaA. EA1 and Sap SLH show the closest structural similarity with an rmsd value of 1.02 Å. Interestingly, we also found that SpaA shows a remarkably similar fold to the *B. anthracis* SLH, suggesting a conserved binding mode to the SCWP. We have included this information in the discussion.

- Line 220 – could you show a map vs model FSC for the D3-D6 fitted into the cryoEM map? How close is it to the reported resolution?

Author response:

The FSC (model-map) is shown here:

Based on the FSC model, the estimated resolution at 0.5 FSC cutoff is 7.7Å. which differs 1.1 Å from the estimated resolution based on the map (6.61 Å)

- Lines 248-249 – the mutants suggest that the X-ray structure lattice is a “true contact” but it does serve as a direct proof. Also, please explain briefly in the text what are these mutants ($\Delta D6$ and N614W-A484Q) since this is the first time they are mentioned in the text.

Author response:

We have now included a brief explanation of the rationalization of these mutants in the main text, and tone down the conclusion of the experiment. Whilst individually they do not provide proof that the X-ray structure captures, at least in part, the native S-layer contacts, our combined results do.

- What happens when the Sap Nbs were mixed with EA1 Nbs? Cell growth should be decreased as the expression of either S-layer proteins cannot compensate for depolymerization.

Author response:

We now more elaboratively assessed the possibly biological effects of Nb-induced loss of S-layer integrity. See our response to Point 1 of Reviewer 1.

- Has any Nbs mixture performed better than others at preventing EA1 polymerisation or were single Nbs and mixtures equally potent?

Author response:

In vitro we have not observed a measurable differences regarding the depolymerization activity of the EA1 nanobodies. In the revised manuscript, we assessed the biological activity of Nb632, Nb633 and Nb643 individually. *In vivo* we observe a clear growth inhibitory effect when using Nb643 during low osmotic conditions. However, cells treated with Nb632 or Nb633 showed no measurable growth effect, despite their efficient depolymerization of isolated S-layers *in vitro*. Based on the structural insights from the crystallization of the

nanobodies nb632 and Nb643, we propose that this disparity arises from the varying accessibility of the epitope. In vitro, conditions provide access to the edges of the S-layer, whereas on the cellular surface, the S-layer adopts a continuous structure accessible only from the top. The structures show that Nb632 binds to the interface of D2 with D5, which is buried in the S-layer, whereas Nb643 bind the hinge region of D2 and D1, which remains accessible in the S-layer, albeit in a different relative conformation. We suspect that the Nb632 epitope is not readily reached in mature, cell-surface localized S-layers, whereas the Nb643 epitope remains accessible, and Nb binding results in an allosteric destabilization of the S-layer. Remarkably, Nb692, a nanobody previously reported to result in in vivo depolymerization of the Sap S-layer binds the D1 – D2 interface in a very similar manner to Nb643 in EA1 (Fiorvanti et al. 2019). Together, these observations indicate that the D1-D2 hinge is an important stabilizing element of the Sap and EA1 S-layers.

- Line 391 – why was the C-terminal domain truncated in the Δ D1 mutant? It is actually a Δ D1 Δ D6 mutant. Please correct this in the text.

Author response:

The C-terminal was not truncated in the Δ D1 mutant. We have now corrected this information in the main text. Thank you for pointing this out.

- Line 541 – please mention that anisotropic correction was performed to obtain the X-ray crystal structure. If this is not the case, why was Staraniso used?

Author response:

The crystals presented a thin plate shape that upon diffraction showed anisotropy. We have now added this information to the manuscript.

Minor

comments

General – doi links of several papers is incorrect (Couture-Tosi, Domanska, Doran, etc), please correct these.

Author response:

We have now updated the reference style to Nature, which does not include doi. All references have been manually checked.

- Line 171 – please add a few words on SbsB for non-specialists.

Author response:

We have now added a few words on SbsB in the main text.

- Were the authors able to identify where Nb633 binds EA1?

Author response:

We could not identify where Nb633 binds. We successfully produce and purified D1, D5 and D6, but did not observe any binding signal to Nb633 when using BLI. However, this does not rule out that Nb633 does not interact with these domains as it could bind the interdomain linker and/or at the interface between two different domains. On the other hand, D2, D3 and D4 were insoluble when overexpressed in *E. coli*.

- Line 228 – Supplementary figure 11 mentions the cysteine mutants, however they are presented later. I would remove any mention until the relevant part of the paper.

Author response:

We have removed the reference to Supplementary Figure 11 in this section and only mention it when we discuss the Cys mutants.

- Line 111 – how does the predicted alphafold EA1AD compare with the experimentally determined structure? Please add a superposition of both to supplementary figure 1.

Author response:

The predicted structure of EA1AD from Alphafold shows a poor global fit with the X-ray structure, as indicated by a high root-mean-square deviation (rmsd) value of 51.88 Å. While the predicted domain structures of D1, D3, D4 and D5 give a reasonably good match to the X-ray structure (Supplementary Fig. 1), Alphafold fails to give a high confidence and accurate prediction of D2, D6 and the relative domain orientation. We have included a superposition in the supplementary Figure 1 as suggested.

- Line 96 – also cite (von Kügelgen et al., 2020).

Author response:

We have now included this reference.

- Line 119 – EA1FL (I assume FL = full-length) should be abbreviated.

Author response:

We have now included this information in the main text.

- Line 120,130 – does recombinant EA1 refer to the EA1FL or EA1AD or both? Please clarify.

Author response:

Both AD and FL form a gel-like pellet consisting of a conglomerate of 2D sheets, we have now specified this in the text.

- Line 124 – please correct “materials and methods” to “methods”.

Author response:

We have now corrected this in the text.

- Line 147 – missing) after Nb11.

Author response:

We have now corrected this in the text.

- Lines 231-232 – add abbreviation to BR, BL, R in the text as well.

Author response:

We have now included these abbreviations in the text.

- Line 257 – typo, should read SC not SD.

Author response:

We have now corrected this in the text.

- Line 276 – does CC refer to map vs model? Please clarify.

Author response:

CC refers to the correlation coefficient of map vs. model, calculated using ChimeraX. We have included a few words to clarify this information.

- Line 279 – DLS, abbreviate.

Author response:

We have included this abbreviation.

- Lines 299-365 – should be Sap not SAP.

Author response:

We have corrected this in the text.

- Line 338 – typo, should be heteromorphous.

Author response:

We have now corrected this typo in the main text.

- Line 555 – was calcium added to the purified EA1 to facilitate polymerization?

Author response:

We did not add calcium during the purification procedure nor before crystallization.

Figures

- Figure 3 and Supplementary figure 9 – please indicate where the interacting loops with the bound calcium are present in the lattice. Do they make any connections to the neighbouring subunits? In Supplementary figure 9 it might be better to move/make transparent the 74.29 Å label to better visualize the contacts made by D6 with its surrounding subunits.

Author response:

Calcium-interacting loops do not establish intermolecular contacts with other monomers of the lattice. However, they enable EA1 to attain its assembly-competent fold by directing the relative orientation of D1-D6. For instance, IL-4 in D6 directly contributes to the R-shape as it interacts with D3 of the same subunit. We have included in the text this information. In supplementary figure 9, we have now moved the vector lines to the side to more clearly show the ribbon representation of the lattice.

- Figure 4 – please show the MC atoms that are involved in H-bond formation in all interfaces presented.

Author response:

We have now modified Figure 4 to show the main chain atoms that are involved in H-bonding.

- Supplementary figure 1 – could the authors show the alphafold prediction for Sap as well? How does the confidence score and prediction look like for Sap?

Author response:

We have now incorporated the AlphaFold prediction of SapAD in Supplementary Figure 1. The individual Ig domains exhibit highly confident predictions, with a pLDDT score exceeding 90. However, the confidence and accuracy in predicting the relative orientation of the D1-D6 are low, similar to what is seen for EA1 (Supplementary Fig. 1).

- Supplementary figure 2 – in a) please correct to “(G and S respectively as indicated on the figure)”.

Author response:

We have now corrected this in supplementary Figure 2.

- Supplementary figure 4 – CDR should be abbreviated and mentioned in the text.

Author response:

We have now included the CDR abbreviation in the figure legend of supplementary Figure 4.

- Supplementary figure 7 – b) the superposition is not very informative, perhaps add transparency to each domain to visualize the similarities better. d) It would be interesting to superpose EA1 and Sap to view their overall arrangement.

Author response:

We have removed supplementary figure 7b and added the superposition of EA1AD and SapAD. As observed, they align poorly (rmsd of 35.18 Å) with the only alignment region being the D1-D2 region.

- Supplementary figure 11 – how do the Cys mutants' lattices compare with the AD lattice? Perhaps adding Fourier transform of the three double mutant alongside AD would be useful.

Author response:

We have measured the lattice parameter for the cysteine mutants and have included this information in section c. (supplementary figure 11). The lattice parameters for the three mutants are the same, being $\alpha= 71 \text{ \AA}$, $\beta= 84 \text{ \AA}$, $\gamma= 107^\circ$, which closely aligns with the values obtained from Cryo-EM. We propose that the slight variation observed could potentially be attributed to the dehydration caused by the staining procedure involving uranyl acetate.

Reviewers' Comments:

Reviewer #1:

Remarks to the Author:

The authors performed the suggested experiment, invalidating the hypothesis made and performed other experiment leading to interesting result (the disruption of the EA1 S-layer with nanobodies compromises the ability of the cell envelope to resist turgor pressure). This improve the manuscript and I recommend publication.

Reviewer #2:

Remarks to the Author:

Structure of the EA1 surface layer of *Bacillus anthracis*

If the authors can add text changes to my points highlighted below, I support publication of this manuscript.

Major comment

Supplementary figure 14 (gallery of negative staining TEM images) – I cannot that say I see a substantial difference between the treatment and the control. For example in the 3rd PBS treated image and the last Nbs image the S-layer looks roughly the same to me. Not sure if these serve as good evidence to show that the S-layer is clearly disrupted. On the balance of the entire manuscript, I would give the authors the benefit of the doubt. However, it is important that the authors point out quite clearly that the negative staining images are ambiguous, and the absolute proof of in vivo disruption is pending.

Minor comments

Line 377 – HPI not Hpi.

Supplementary figure 19 legend – EA1 not Ea1.

Line 643 – liquid ethane temp 180K would be that the grids are not vitreous. Probably the temperature was lower? Please correct.

Figures

Might be good to add the map vs model FSC to the cryoEM analysis (Supplementary figure 10).

REVIEWERS' COMMENTS

Reviewer #1 (Remarks to the Author):

The authors performed the suggested experiment, invalidating the hypothesis made and performed other experiment leading to interesting result (the disruption of the EA1 S-layer with nanobodies compromises the ability of the cell envelope to resist turgor pressure). This improve the manuscript and I recommend publication.

We thank reviewer #1 for his/her time and advice that allowed us to improve the manuscript.

Reviewer #2 (Remarks to the Author):

Structure of the EA1 surface layer of Bacillus anthracis

If the authors can add text changes to my points highlighted below, I support publication of this manuscript.

We thank reviewer #2 for his/her time and advice that allowed us to improve the manuscript.

Major comment

Supplementary figure 14 (gallery of negative staining TEM images) – I cannot that say I see a substantial difference between the treatment and the control. For example in the 3rd PBS treated image and the last Nbs image the S-layer looks roughly the same to me. Not sure if these serve as good evidence to show that the S-layer is clearly disrupted. On the balance of the entire manuscript, I would give the authors the benefit of the doubt. However, it is important that the authors point out quite clearly that the negative staining images are ambiguous, and the absolute proof of in vivo disruption is pending.

We have now added a statement in the main text that reflects the reviewer's concern regarding the ambiguity of the TEM images.

Minor comments

Line 377 – HPI not Hpi.

We have corrected this in the main text

Supplementary figure 19 legend – EA1 not Ea1.

We have corrected this in the main text

Line 643 – liquid ethane temp 180K would be that the grids are not vitreous. Probably the temperature was lower? Please correct.

We have corrected this in the main text

Figures

Might be good to add the map vs model FSC to the cryoEM analysis (Supplementary figure 10).
We have added the map vs model FSC to Supplementary Figure 10.